



# Hemispheric contrasts in ice formation in stratiform mixed-phase clouds: Disentangling the role of aerosol and dynamics with ground-based remote sensing

Martin Radenz[1], Johannes Bühl[1], Patric Seifert[1], Holger Baars[1], Ronny Engelmann[1], Boris Barja González[2], Rodanthi-Elisabeth Mamouri[3,4], Félix Zamorano[2], and Albert Ansmann[1]

[1]Leibniz Institute for Tropospheric Research (TROPOS), Leipzig, Germany
[2]Atmospheric Research Laboratory, University of Magallanes, Punta Arenas, Chile
[3]Department of Civil Engineering and Geomatics, Cyprus University of Technology of Technology, Limassol, Cyprus
[4]ERATOSTHENES Centre of Excellence, Limassol, Cyprus

**Correspondence:** Martin Radenz (radenz@tropos.de)

**Abstract.**

Multi-year ground-based remote-sensing datasets acquired with the Leipzig Aerosol and Cloud Remote Observations System (LACROS) at three sites: a highly polluted central European site (Leipzig, Germany), a polluted and strongly dust-influenced eastern Mediterranean site (Limassol, Cyprus), and a clean marine site in the southern mid-latitudes (Punta Arenas, Chile) are

5 used to contrast ice formation in shallow stratiform liquid clouds. These unique, long-term datasets at key sites of aerosol-cloud interaction provide a deeper insight into cloud microphysics. The influence of temperature, aerosol load, boundary-layer coupling and gravity wave motion on ice formation is investigated. With respect to previous studies of regional contrasts in the properties of mixed-phase clouds our study contributes the following new aspects: (1) Sampling aerosol optical parameters as a function of temperature, the average backscatter coefficient at supercooled temperatures is within a factor of 3 at all three sites.

(2) Ice formation was found to be more frequent for cloud layers with cloud top temperatures above $-15\,^{\circ}\mathrm{C}$ than indicated by prior lidar-only studies at all sites. A virtual lidar-detection threshold of IWC needs to be considered in order to bring radar-lidar-based studies in agreement with lidar-only studies. (3) At similar temperatures, cloud layers which are coupled to the aerosol-laden boundary layer show more intense ice formation than de-coupled clouds. (4) Liquid layers formed by gravity waves were found to bias the phase occurrence statistics below $-15\,^{\circ}\mathrm{C}$. By applying a novel gravity wave detection approach

using vertical velocity observations within the liquid-dominated cloud top, wave clouds can be classified and excluded from the statistics. After considering boundary layer and gravity-wave influences, Punta Arenas shows lower fractions of ice containing clouds by $0.1$ to $0.4$ absolute difference at temperatures between $-24$ and $-8\,^{\circ}\mathrm{C}$. These differences are potentially caused by the contrast in the INP reservoir between the different sites.

*Copyright statement.* TEXT



## 1   Introduction

Clouds and aerosol are inseparably coupled, linked via complex pathways of interaction whose outcome manifests in the macroscopic properties of precipitation and radiation fields. On the one hand, aerosol particles are required as cloud condensation nuclei on which cloud droplets can form. On the other hand, primary ice formation in the heterogeneous freezing temperature range from 0 to approximately -40°C requires ice nucleating particles (INP) to be present in the aerosol reservoir. The ways in which aerosol and cloud particles interact are controlled by the dynamics and thermodynamics of the atmospheric environment. Thermodynamic processes are considered to dominate the cloud microphysical properties because they control the amount of water vapour that is available for being transferred to either the liquid or the ice phase. This dominance makes it difficult to isolate aerosol-related effects in observations of cloud properties.

Nevertheless, observations as well as aerosol-permitting model studies suggest a considerable influence of the aerosol conditions on the properties and evolution of clouds and precipitation (Seifert et al., 2012; Possner et al., 2017; Solomon et al., 2018; Zhang et al., 2018). Solomon et al. (2018) used high-resolution modelling of Arctic mixed-phase clouds to show, that perturbations in the INP concentration dominate over changes in the cloud condensation nuclei (CCN) concentrations. Cloud chamber studies suggest, that holding CCN constant, the ratio of ice to liquid water content in the steady state is predominantly controlled by INP concentrations (Desai et al., 2019).

There is also a distinct spatio-temporal variability of the performance of weather and climate model simulations, which is attributed to the insufficient representation of aerosol-cloud-dynamics interaction processes in the models (Fan et al., 2016; Seinfeld et al., 2016). For instance, the reasons for the less accurate treatment of the radiative balance in the southern-hemispheric mid-latitudes compared to their northern-hemispheric counterpart are still debated (Trenberth and Fasullo, 2010; Grise et al., 2015). The atmosphere of the southern hemisphere's mid-latitudes is a unique component of the Earth's climate system. It's the stormiest (e.g. Young, 1999) and one of the cloudiest places on Earth (Haynes et al., 2011; Naud et al., 2014), but process understanding of clouds in that region is still limited. The reported biases in the solar radiation budget are attributed to shallow supercooled liquid topped clouds, which are insufficiently represented by current models (Bodas-Salcedo et al., 2014; Kay et al., 2016; Bodas-Salcedo et al., 2016; Kuma et al., 2020). These radiation biases affect estimates of sea surface temperature and surface precipitation, ultimately affecting the energy balance at the surface (Franklin et al., 2013; Hyder et al., 2018). There is an ongoing controversy about the reasons for the observed differences and prevailing model deficiencies, but indications are given that a combination of hemispheric contrasts in aerosol load and atmospheric dynamics plays a role.

Different causes for the excess of the supercooled liquid cloud layers are proposed. On the one hand, reason for the excess of supercooled liquid water in southern-hemispheric cloud systems could be the inhibition of ice formation caused by the lack of ice nucleating particles in the predominantly pristine environment of the Southern Ocean (Hamilton et al., 2014), where terrestrial sources, which are frequently considered a good source for INP, are rare or far apart (Vergara-Temprado et al., 2017). In the heterogeneous freezing regime, suitable aerosol particles are a prerequisite for ice formation and missing ice formation as a sink for cloud water, the liquid phase may be sustained for long periods of time. On the other hand, dynamical processes could lead to an enhancement of supercooled liquid water. Korolev (2007) showed, that depending on the number and size of



ice crystals, a threshold vertical velocity can be found, which allows for sufficient supersaturation to grow the ice as well as the
liquid phase. Gravity waves have been suggested to play a role in the phase partitioning of Southern Ocean clouds (Alexander
et al., 2017; Silber et al., 2020). Due to the orographic effects and the strong westerlies, gravity waves are a general feature in
the vicinity of all landmasses in the middle and high latitudes of the southern hemisphere (Sato et al., 2012; Alexander et al.,
2016). It is however also noteworthy that stronger turbulence increases the amount of ice formed in stratiform cloud layers
(Bühl et al., 2019). Indications are thus given that it is necessary to also consider turbulence in studies of ice formation.

The undetermined contributions of dynamical and aerosol effects on the observed excess of supercooled liquid in the south-
ern hemisphere requires dedicated attribution studies. In numerous previous studies, liquid-topped supercooled stratiform cloud
layers have been proven to be suitable natural laboratories for the investigation of the relationships between aerosol properties,
thermodynamics and microphysical properties of clouds in the heterogeneous freezing regime. Temperature at which the ice
formation occurs, needs to be strongly constrained, because the concentration of efficient INP increases rapidly with decreasing
temperature (e.g. Kanji et al., 2017). Accordingly, the amount of ice formed also increases for lower temperatures (Bühl et al.,
2016). The lowest temperature in such clouds occurs on top of the liquid-dominated layer, hence the cloud top temperature
(CTT) can be used to constrain the ice formation temperature. Turbulence is usually confined to the liquid-dominated cloud top
(Westbrook and Illingworth, 2013; de Boer et al., 2009) and due to the limited thickness of this layer, secondary ice formation
or ice multiplication are strongly constrained (Fukuta and Takahashi, 1999; Myagkov et al., 2016). Contrasting microphysical
properties observed in pristine, clean regions with observations from areas with higher aerosol load, e.g., allows to advance
understanding of the impact of different aerosol loads (Choi et al., 2010; Kanitz et al., 2011; Seifert et al., 2015; Tan et al.,
2014). Recent studies based on the A-Train satellite constellation suggest systematically lower ice amounts in the southern
mid-latitudes (Zhang et al., 2018) and a strong susceptibility to dust load (Villanueva et al., 2020). Supercooled liquid clouds
are frequent over the Southern Ocean (Huang et al., 2015; Hu et al., 2010) and studies by Kanitz et al. (2011) and Choi et al.
(2010) showed, that - at similar temperatures - ice is formed less frequently by liquid layers in the southern hemisphere mid-
latitudes, than in the northern hemisphere. The study of Kanitz et al. (2011) first used a ground-based lidar at Punta Arenas
(53.1°S 70.9°W, Chile) to asses the thermodynamic phase of stratiform mixed phase clouds above the Southern Hemisphere
mid-latitudes. Major caveats of this study are the limited duration of the observations during austral summer and the limitations
of the lidar-only setup. Alexander and Protat (2018), using ground-based lidar and A-Train from Cape Grim (40.7°S 144.7°E,
Australia) confirm the basic findings also for the eastern parts of the Southern Ocean. This study, similarly to McErlich et al.
(2021), emphasizes the problems in A-Train derived datasets detecting shallow clouds in the lowermost part of the atmosphere.

Recent activities include shipborne, land-based and aircraft campaigns targeting aerosols and clouds above the Southern
Ocean between Australia and Antarctica (60 to 160°E). An overview is provided by McFarquhar et al. (2020). In terms of
stratiform clouds, the large abundance of supercooled liquid water, occasionally down to −30°C, was confirmed. But, apart
the year-long lidar/radar dataset at Macquarie Island (54.6°S 158.9°E, Australia) the cloud observations focused on austral
summer.

Using a shiporne dataset, Mace and Protat (2018) also found frequent liquid-dominated clouds with low radar reflectivities
and 1/3 of the liquid layers only observed with lidar. Comparing the observations with a Cloud-Aerosol Lidar and Infrared





Pathfinder Satellite Observations (CALIPSO) dataset from Hu et al. (2010), they found an overestimation of supercooled
liquid in the satellite dataset, especially strong at temperatures above $-15°$C. In a follow-up study, Mace et al. (2020) refined
the CALIPSO classification scheme, leading to more frequent detections of the mixed phase, especially during wintertime and
in the lower latitudes of the Southern Ocean. However, no CTT-resolved phase occurrence statistics is presented. Liquid layers
in deeper clouds, observed during another shipborne campaign (McFarquhar et al., 2020; Alexander et al., 2021), could only be
reproduced in regional model simulations, when INP parametrization was tuned to lower concentrations (Vignon et al., 2021).
Zaremba et al. (2020) investigated airborne active remote sensing observations of Southern Ocean clouds south of Tasmania.
They also found widespread liquid cloud tops at temperatures down to $-30°$C. By investigating the ground-based remote
sensing dataset assembled at McMurdo (77.8°S 166.7°E, Antarctica), Silber et al. (2018) found frequent long-lived liquid
topped clouds, also below $-30°$C.

Yet, a statistical analysis of the relationship between both, aerosol conditions, cloud vertical dynamics, and the phase par-
titioning in stratiform cloud layers of the southern-hemisphere mid-latitudes based on long-term observations was not estab-
lished. One reason is, that, despite increased activity in the recent past, ground-based remote sensing observations of clouds
and aerosol are still sparsely distributed in the Southern Ocean and at the coast of Antarctica.

Goal of this study is to analyze long-term ground-based remote sensing observations of aerosol properties, cloud micro-
physics and atmospheric dynamics from three sites with strongly contrasting aerosol conditions in order to attribute ice forma-
tion in the heterogeneous freezing regime to atmospheric dynamics and aerosol conditions. For that attribution approach, we
utilized the recent campaigns of the Leipzig Aerosol and Cloud Remote Observations System (LACROS) at Leipzig (51.4°N
12.4°E, Germany), Limassol (34.7°N 33.0°E, Cyprus) and Punta Arenas (53.1°S 70.9°W, Chile) which provide datasets, that
cover the aerosol conditions of a continental northern hemispheric background site, a hot-spot of mineral dust, and the marine-
dominated pristine Southern Ocean, respectively. Hence, these datasets collected with a single set of ground-based remote
sensing instrumentation provide an ideal basis for contrasting studies. The broad variety of instruments covers the decisive
properties of aerosols, dynamics, clouds and precipitation for a more comprehensive picture of aerosol-cloud interaction. The
observations at Punta Arenas provide the first multi-year dataset of synergistic ground-based remote sensing observations in
the western half of the Southern Ocean and allow to contextualize prior findings.

The paper is structured as follows: In section 2 the instrumentation and campaigns are described (Sec. 2.1), followed by
the retrievals of aerosol properties (Sec. 2.2) and the synergystic Cloudnet retrieval (Sec. 2.3). Afterwards, the methods for
cloud selection and vertical velocity characterization are introduced. Instruments, field campaigns and synergistic retrievals are
described first. Section 2 covers the methods, consisting of the retrieval of the aerosol statistics, the automated cloud selection
algorithm (Sec. 2.4) and the characterization of vertical velocity (Sec. 2.5). The statistics is presented in Sec. 3, including the
lidar-derived average profiles of aerosol optical properties (Sec. 3.1) and the cloud phase statistics with special emphasis on
instrument sensitivity (Sec. 3.2.1). Boundary layer coupling (Sec. 3.2.2) and vertical dynamics are discussed in Sec. 3.2.3.
The amount and efficiency of ice production is assessed in Sec. 3.2.4. The study concludes with a discussion of the contrast
identified in aerosol load and stratiform cloud properties (Sec. 4) followed by a summary and outlook (Sec. 5).





**Table 1.** Specifications of the LACROS instruments used in this study.

| Instrument (Reference) | Frequency $\nu$ Wavelength $\lambda$ | Measured quantity | Temporal resolution | Vertical range | Vertical resolution |
|---|---|---|---|---|---|
| Doppler cloud radar METEK Mira-35 (Görsdorf et al., 2015) | $\nu = 35\,\mathrm{GHz}$ | Radar reflectivity factor | 3.5 s | $150 - 13000\,\mathrm{m}$ | 30 m |
| | | Vertical velocity | 3.5 s | $150 - 13000\,\mathrm{m}$ | 30 m |
| | | Linear depolarization ratio | 3.5 s | $150 - 13000\,\mathrm{m}$ | 30 m |
| Raman-Polarization Lidar Polly[XT] (Engelmann et al., 2016) | $\lambda = 355, 532, 1064\,\mathrm{nm}$ | Attenuated backscatter coeff. | 30 s | $100 - 15000\,\mathrm{m}$ | 7.5 m |
| | $\lambda = 355, 532\,\mathrm{nm}$ | Raman backscatter signal | 1 h | $300 - 5000\,\mathrm{m}$ | $\sim 50\,\mathrm{m}$ |
| | $\lambda = 355, 532\,\mathrm{nm}$ | Linear depolarization ratio | 30 s | $100 - 15000\,\mathrm{m}$ | 7.5 m |
| Microwave radiometer RPG HATPRO-G2 (Rose et al., 2005) | $\nu = 22.24 - 31.4\,\mathrm{GHz}$ | Brightness temperatures | 1 s | column integral | |
| | $\nu = 51.0 - 58.0\,\mathrm{GHz}$ | Brightness temperatures | 1 s | column integral | |
| Doppler Lidar Halo Streamline (Pearson et al., 2009) | $\lambda = 1.5\,\mu\mathrm{m}$ | Attenuated backscatter coeff. | 2 s | $48 - 12000\,\mathrm{m}$ | 48 m |
| | | Vertical velocity | 2 s | $48 - 12000\,\mathrm{m}$ | 48 m |
| Ceilometer Jenoptik chm15kx | $\lambda = 1064\,\mathrm{nm}$ | Attenuated backscatter coeff. | 30 s | $15 - 15300\,\mathrm{m}$ | 15 m |
| Optical disdrometer Ott Parsivel[2] (Löffler-Mang and Joss, 2000) | $\lambda = 650\,\mathrm{nm}$ | Hydrometeor size distribution | 30 s | 4 m | - |
| Sun photometer Cimel CE318-T (Barreto et al., 2016) | $\lambda = 340 - 1064\,\mathrm{nm}$ | Aerosol optical thickness | variable | column integral | |

## 2 Data and Methods

This section introduces the datasets and methods used in the remainder of this study. Starting with the campaigns and in-
strumentation (Sec. 2.1), followed by a short description of the retrievals used (Sec. 2.2 and 2.3) and finally the methods for
automated selection of shallow stratiform clouds (Sec. 2.4) and characterization of vertical velocity dynamics (Sec. 2.5).

### 2.1 Datasets from Leipzig, Limassol and Punta Arenas

Basis of the observational datasets presented below is the Leipzig Aerosol and Cloud Remote Observations System (LACROS),
the mobile ground-based remote-sensing supersite of the Leibniz Institute for Tropospheric Research (TROPOS), Leipzig,
Germany. The instrumentation used for the synergistic approaches applied in this study comprises a Mira-35 35 GHz scanning
cloud radar, a Polly[XT] multi-wavelength Raman and depolarization lidar, a Streamline $1.5\,\mu\mathrm{m}$ scanning Doppler lidar, a HAT-
PRO 14-channel microwave radiometer, a $1064\,\mathrm{nm}$ ceilometer, an optical disdrometer and radiation sensors. Main properties
of the sensors are summarized in Table 1.

LACROS was established in 2011 and was first introduced by Bühl et al. (2013b). After the initial setup phase, the basic set
of instrumentation of LACROS has been kept unchanged since the year 2014. Hence, this comparative study uses data from the





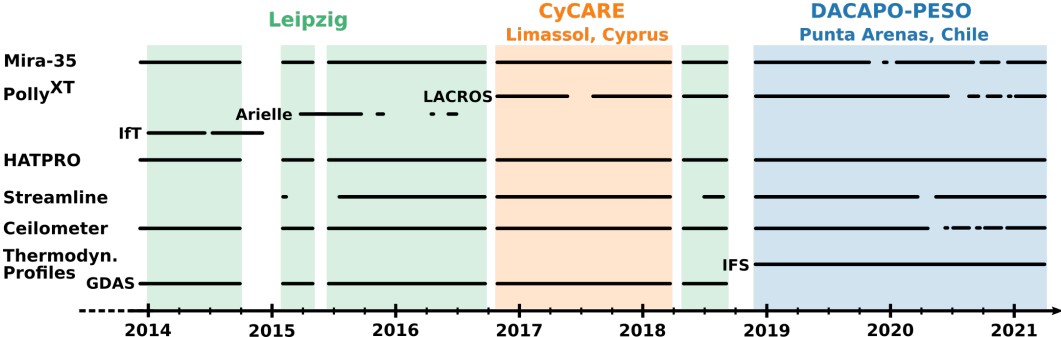

**Figure 1.** Overview on instrument availability and campaign durations of the LACROS deployments covered in this study. Characteristics of the three different Polly[XT] instruments used at Leipzig and named IFT, Arielle, and LACROS are described in Engelmann et al. (2016).

6 year period since then. During that period, LACROS was, besides several 4-8 weeks short-term deployments at other sites, stationed at Leipzig, Limassol, and Punta Arenas.

Fig. 1 provides an overview on the campaigns and instrument availability. From October 2016 to March 2018 LACROS was located at Limassol for the Cyprus Clouds, Aerosol and Rain Experiment (CyCARE). In the context of the present study, this
location in the eastern Mediterranean in the middle of the dust belt serves as a reference for a highly aerosol-laden environment. First results of the CyCARE campaign are described by Ansmann et al. (2019). For the second long term campaign, LACROS was moved to Punta Arenas in November 2018 for the two-year Dynamics, Aerosol, Clouds And Precipitation Observations in the Pristine Environment of the Southern Ocean (DACAPO-PESO) field campaign. Located at 53.1°S, Punta Arenas is in the midst of the Southern Ocean, farther south than any other continental land mass or major island apart Antarctica. Under
prevailing westerly flow, the next land mass upwind is more than $8000\,\mathrm{km}$ away and significantly to the north. Hence, the free tropospheric aerosol is predominantly of marine origin with low concentrations overall. First results of the DACAPO-PESO campaign regarding events of significant aerosol load (Ohneiser et al., 2020; Floutsi et al., 2021), liquid cloud microphysics (Jimenez et al., 2020) and integrated water vapor observations (Bromwich et al., 2020) are already published. Observations at Leipzig were performed before October 2016 and between the deployments to Limassol and Punta Arenas. Due to the location
in central Europe, the Leipzig dataset serves as a reference for continental background in the northern hemisphere.

Data coverage for each of the sites is summarized in Table 2. Gaps in the observations were usually caused by periods of short-term deployments at other sites, failure of primary instruments, or power cuts (see also Fig. 1). Coverage with the core instrumentation is generally well above 80%, only exception being Doppler lidar and Polly[XT] at Leipzig.

## 2.2 Lidar-based aerosol statistics

As a proxy for assessing the aerosol conditions at the three measurement sites, we used vertically resolved observations from the Polly[XT] lidar system (Althausen et al., 2009; Engelmann et al., 2016). Quantities of interest are the aerosol backscatter





**Table 2.** Overview on Location, data availability, climate, aerosol load and related studies for the LACROS datasets used.

| Location | Leipzig, Germany | Limassol, Cyprus | Punta Arenas, Chile |
|---|---|---|---|
| | 51.4°N 12.4°E | 34.7°N 33.0°E | 53.1°S 70.9°W |
| Station altitude | 125 m asl | 11 m asl | 9 m asl |
| Campaign name | | CyCARE | DACAPO-PESO |
| Duration | 976 d | 524 d | 765 d |
| Cloudnet data | 771 d | 520 d | 674 d |
| Doppler lidar data | 513 d | 523 d | 717 d |
| Polly$^{\mathrm{XT}}$ data | 633 d | 460 d | 702 d |
| Climate | Northern mid-latitudes | Northern Tropics | Southern mid-latitudes |
| Typical aerosol load | Continental background, occasionally dust | Dust, marine, continental | Marine, occasionally continental |
| Related studies | Bühl et al. (2016) | Ansmann et al. (2019) | Ohneiser et al. (2020) Bromwich et al. (2020) Jimenez et al. (2020) Floutsi et al. (2021) |

coefficient, the extinction coefficient, the particle linear depolarization ratio, and the cloud-relevant concentration of INP. Their retrieval is explained below.

Basis for the statistical analysis are profiles of particle backscatter coefficient $\beta_{\mathrm{p}}$ at 532 nm wavelength. The profiles are

computed with the Klett method (Fernald, 1984) by the PollyNET retrievals whenever conditions are suitable (Baars et al., 2016, 2017; Yin and Baars, 2021). The PollyNET retrieval chain also ensures a homogenized analysis of the data from the three different Polly$^{\mathrm{XT}}$ instruments, which were utilized in the frame of this study (see Fig. 1 and Engelmann et al., 2016; Baars et al., 2016). Profiles of the particle linear depolarization ratio (hereafter referred to as particle depolarization ratio) are only calculated when the ratio of molecular backscatter coefficient to $\beta_{\mathrm{p}}$ exceeds a value of 18. Additionally, any particle depolarization ratios

larger than 0.7 are masked, as they are indications for noise artifacts in the cross-polarized signal component. All profiles are then filtered with the co-located Cloudnet target classification (see Section 2.3) to exclude clouds, especially optically thin ice clouds only seen by the cloud radar. Finally, a manual screening excluded fragments of thin liquid clouds, which would otherwise artificially increase $\beta_{\mathrm{p}}$. For the averages, the optical data of each retrieved profile is binned to vertical intervals of 200 m or 3 K.

The derived average optical properties can be used to estimate aerosol microphysical properties, such as concentrations of ice nucleating particles (INP). This is an important step in order to evaluate the datasets of the three sites with respect to contrasts in the potential contribution of aerosol effects on heterogeneous ice formation efficiency. Conversion from optical properties as observed by lidar to microphysical aerosol properties is based on the parametrizations described by Mamouri and Ansmann (2016). By means of this approach, the lidar-measured aerosol extinction coefficient is converted to the number and surface

concentration $N_{500}$ and $S_{500}$, respectively, of aerosol particles larger than 500 nm in diameter. These quantities are applied in available in-situ-based parametrizations for the retrieval of INP concentrations. Prerequisite for the retrieval is a correct





aerosol typing, as different types of particles differ by orders of magnitude in their ice forming efficiency. In order to do so, the average backscatter profile is separated into the categories marine, continental and mineral dust, based on air mass source (see Appendix A and Radenz et al., 2021) and particle depolarization ratio (one-step POLIPHON; Mamouri and Ansmann, 2017).

The average extinction is calculated from the profiles of $\beta_{\mathrm{p}}$ by assuming a typical lidar ratio: marine $20\,\mathrm{sr}$, continental $50\,\mathrm{sr}$ and dust $45\,\mathrm{sr}$ (Müller et al., 2007; Baars et al., 2017; Bohlmann et al., 2018). The extinction coefficient is in the next step converted to $N_{500}$ and $S_{500}$, using sun-photometer-based conversion factors (Mamouri and Ansmann, 2016). Then, the above mentioned INP parametrizations are applied for each aerosol class, such as DeMott et al. (2015) for mineral dust, DeMott et al. (2010) for continental aerosol or McCluskey et al. (2018b, a) for marine aerosol.

## 185 2.3 Cloudnet processing

For determination of cloud macro- and microphysical properties and as basis for the stratiform cloud identification synergies between lidar, cloud radar, microwave radiometer and meteorological data are utilized. State-of-the-art routines for achieving this requirement are comprised in the Cloudnet retrieval (Illingworth et al., 2007). Cloudnet re-grids the observations to a common resolution ($30\,\mathrm{s}$ and $31.18\,\mathrm{m}$, determined by the vertical resolution of Mira-35 with a pulse length of $208\,\mathrm{ns}$) and

provides products, such as a target classification and retrievals of microphysical cloud properties. Regular scans of the MIRA-35 cloud radar (hourly) and the Streamline Doppler lidar (twice per hour) are not used within the Cloudnet processing scheme. Profiles of temperature, pressure and humidity are obtained from the ECMWF's IFS analysis for Punta Arenas and GDAS analysis for Leipzig and Limassol. Liquid water path (LWP) and integrated water vapor (IWV) are retrieved from brightness temperature observations of the microwave radiometer in two frequency bands from $22.24$ to $31.4\,\mathrm{GHz}$ and $51.0$ to $58.0\,\mathrm{GHz}$

(7 channels each). The statistical retrieval is based on long-term radiosonde observations (Leipzig and Limassol) and high-resolution reanalysis data (Punta Arenas). Attenuated backscatter of the ceilometer is regularly cross-calibrated with Polly$^{\mathrm{XT}}$ using the calibrated attenuated backscatter of PollyNET. Gaps in the ceilometer data are filled with Polly$^{\mathrm{XT}}$ observations.

## 2.4 Automated cloud selection and characterization

While Cloudnet provides a concise overview on the macrophysical properties of the observed cloud layers, it requires dedicated,

automatic and reproducible filtering algorithms for the selection of the targeted stratiform, supercooled cloud systems. Based on the data cube LARDA[3] (Bühl et al., 2021), the approach of Bühl et al. (2016) is implemented into an automated selection algorithm.

An example of the Cloudnet processing of measurement data and the application of the cloud selection scheme is shown in Fig. 2. Starting with a profile of the Cloudnet target classification mask (Hogan and O'Connor, 2004), consecutive pixels

classified as cloud pixels (liquid droplets, ice, ice and supercooled droplets) are grouped together and defined as features. Then, single features in neighboring profiles are connected to coherent cloud objects, if similar types of hydrometeors were observed in matching heights. For the analysis, the cloud objects are filtered for shallow stratiform clouds, which are liquid topped and either have an ice virga or not (rectangles in Fig. 2d). An overview of the microphysical parameters sampled for each cloud object is provided in Table 3. It is assumed, that if ice is formed in an liquid layer it also sediments out of the cloud.





This is required, as the signal in the top layer is dominated by return from liquid droplets and Cloudnet provides no reliable mixed-phase classification there.

To pinpoint potential effects of aerosol load, thermodynamic and dynamic drivers of ice formation have to be constrained. This is especially important, as the sites are situated in different climate zones. We presume for our study, that thin stratiform clouds serve as a natural laboratory with only a limited number of microphysical processes being possible.

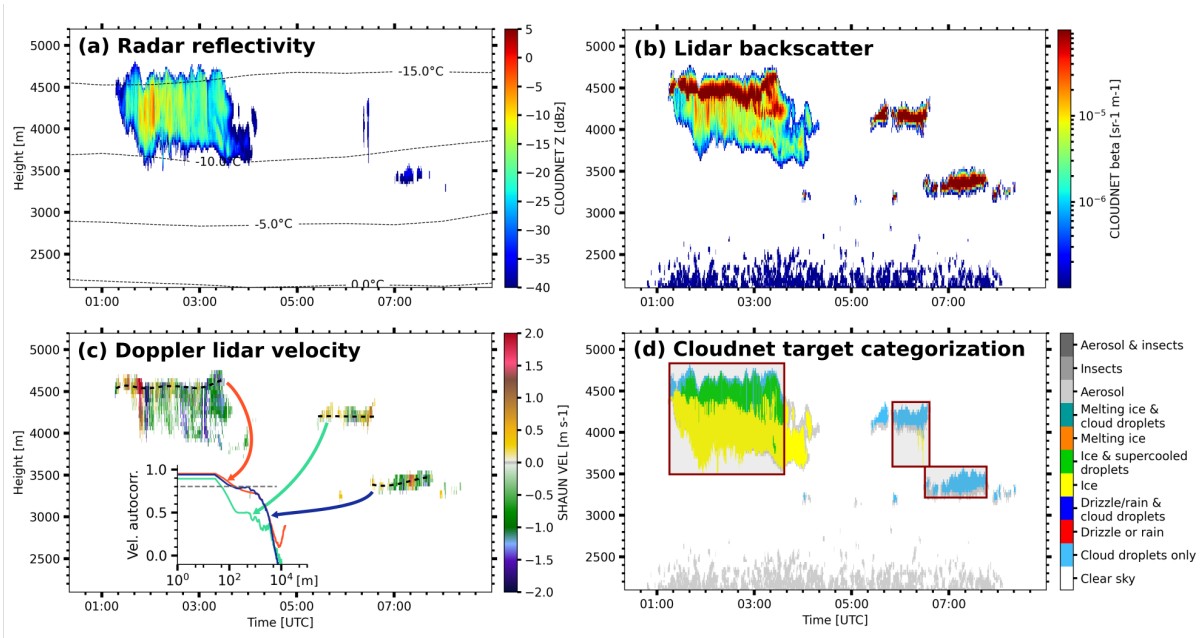

**Figure 2.** MIRA-35 radar reflectivity overlaid with the temperature (a), PollyXT attenuated backscatter (b), Streamline Doppler lidar vertical velocity (c), and Cloudnet target categorization (d) on the 28 November 2018 between 00:20 and 09:00 UTC. Red rectangles in (d) show the detected liquid-topped shallow stratiform clouds. The inset in (c) shows the autocorrelation function of the vertical velocities in the liquid-dominated layer (heights indicated by dashed lines) of each cloud. Further detailes are provided in Sec. 2.5.

Clouds driven by convective processes are excluded by only allowing clouds that were observed for more than 20 minutes and showed a smooth cloud top height (standard deviation $< 150\,\mathrm{m}$). The average thickness of the liquid layer has to be less than $350\,\mathrm{m}$. This criterion is motivated by Fukuta and Takahashi (1999) and excludes secondary ice formation processes and riming (see also Myagkov et al., 2016). Seeding by ice clouds above is avoided by excluding clouds with ice pixel above the liquid layer. Only cloud objects with top temperatures between $-38$ and $0°C$ are considered for the statistics. For the

phase occurrence frequency statistics, a cloud is classified as ice producing, if ice pixel were observed $180\,\mathrm{m}$ below the liquid-dominated layer within at least $5\%$ of the duration. On the other hand, cloud objects are classified as liquid-only, if ice pixel are observed to sediment out of the liquid-dominated layer less than $5\%$ of the time. This threshold is simpler, than the decision tree used by Bühl et al. (2013a), but provides similar results. The resulting fraction is weighted by the length of the cloud. Similarly to (Bühl et al., 2016), the properties of the ice in the virga are taken $180\,\mathrm{m}$ below the liquid-dominated cloud top





**Table 3.** Parameters sampled from the cloud objects. A more detailed description is provided in Bühl et al. (2016).

| Parameter | Time series | Description |
|---|---|---|
| Cloud top temperature | yes | Temperature of the topmost liquid pixel |
| Cloud top height | yes | Geometrical height of the topmost liquid pixel |
| Liquid layer thickness | yes | Geometrical depth of the liquid-dominated cloud top layer |
| Phase classification | no | Ice pixel observed below liquid-dominated layer $> 5\%$ of duration |
| Cloud base vertical velocity | yes | Doppler lidar vertical velocity at the base of the liquid-dominated layer |
| Liquid water content | yes | Liquid water content of the liquid-dominated layer |
| Reflectivity virga | yes | Radar reflectivity $180\,\mathrm{m}$ below the base of the liquid-dominated layer |
| Ice water content in virga | yes | Reflectivity derived IWC $180\,\mathrm{m}$ below the base of the liquid-dominated layer |
| Ice-to-liquid content ratio | yes | Ratio of IWC virga to LWC |

layer to avoid uncertainties in the liquid base estimate and sublimation within the virga. The ice water content (IWC) is derived from the radar reflectivity and the temperature using the Hogan et al. (2006) retrieval. With this automated cloud selection scheme multi-year datasets can be analyzed using objective criteria, while yielding statistics similar to manual cloud selection (e.g. Seifert et al., 2010; Kanitz et al., 2011; Seifert et al., 2015).

## 2.5   Gravity wave detection

In order to enable the attribution of aerosol and dynamical effects on the phase partitioning in the stratiform cloud dataset, an approach is required to assign cloud dynamics regimes to each cloud object. Here we focus on the temporal structure of vertical velocity, to constrain the dynamics forcing on a cloud. Usually, shallow clouds are characterized by a fully developed turbulence in the liquid-dominated cloud top (Bühl et al., 2019), where the vertical motion is driven by cloud top cooling (e.g. Shao et al., 1997; Fang et al., 2014; Simmel et al., 2015). In the turbulent layer at cloud top, up- and downdrafts alternate at

horizontal scales in the order of $100\,\mathrm{m}$ or less.

     However, orographic gravity waves can trigger vertical motion and associated cloud formation, as well. Microphysical processes in these wave clouds are governed by large-scale dynamics, where vigorous up- and downdrafts may appear stationary. Due to this dynamics, the mixed-phase and the ice phase are horizontally separated, with the liquid drops predominantly in the ascending branch and the ice particles in the descending branch (Heymsfield and Miloshevich, 1993; Baker and Lawson,

2006). The properties of the horizontal wind field determines the regions of the up- and downdraft in such orographic clouds. Observing these clouds by stationary ground-based remote sensing might thus not sample the full horizontal extend of the cloud, which causes a misclassification of the cloud in terms of liquid-only and ice-producing.

     Also, the flow is highly laminar, opposed to the confined, fully developed turbulence found in layered mixed-phase clouds. Cloud microphysics in wave conditions cannot directly be compared to layered clouds. Ice formation in those clouds is dom-

inated by homogeneous and evaporation freezing (Cotton and Field, 2002), which are not subject to this study and should be





excluded from our statistics. On the other hand, the frequent occurrence of atmospheric gravity waves in a specific region might increase the frequency of thermodynamic conditions that favor the presence of a sustained liquid phase. As Korolev (2007) demonstrate, long-lasting steady updrafts are required in order to make the liquid phase dominating over the ice phase. Probing an observational dataset for the presence of long-lasting updrafts could therefore provide a hint on the role of atmospheric

gravity waves in the occurrence of enhanced concentration of supercooled liquid.

Turbulence properties of the liquid layer in the clouds subject to our study can be derived from the vertical velocity observation by Doppler lidar (Bühl et al., 2019). The small size of droplets in the liquid-dominated cloud-top layer and their negligible terminal velocity makes them tracers of air motion. From the Doppler lidar observations, the vertical velocity is sampled at the pixel with the maximum backscatter out of the heights identified as liquid-containing in the Cloudnet classification. The height

of this sampling is indicated as dashed lines in Fig. 2c. The temporal resolution of the resulting time series is $2\,\mathrm{s}$, equally to the Doppler lidar raw data. This time-series is then used to get insights into the turbulent properties of the cloud top layer.

The structure of the vertical velocity timeseries can be characterized with the autocorrelation function

$$\Psi(\tau) = \sum_{t} v_t\, v_{t+\tau} \tag{1}$$

with the temporal shift $\tau$ and the vertical velocity $v$ at time $t$. To compare different cloud objects, the autocorrelation function

$\Psi(\tau)$ is normalized with $\Psi(0)$. The temporal shift $\tau$ from the observations is converted into a horizontal shift or autocorrelation length $l$ with the Cloudnet model-based horizontal wind velocity $v_{\mathrm{hor}}$:

$$\Psi(l) = \Psi(\tau)\, v_{\mathrm{hor}} \tag{2}$$

Similarly the vertical velocity spectral power density is calculated by a Fast Fourier Transform of the vertical velocity time series. High autocorrelation coefficients for large shifts and low power density are indications for wave-driven, low turbulent

flow. The inset in Fig. 2c shows the autocorrelation function for each of the identified cloud objects. All of them are weakly affected by gravity waves, but small scale turbulence dominates. When the cloud phase is investigated for an influence of gravity waves (Sec. 3.2.3), the length scale, where the autocorrelation drops below $0.8$ is used to separate wave driven clouds from clouds with fully developed turbulence.

## 3   Results

In this section, the thermodynamic phase partitioning and quantitative ice mass production in ice forming shallow cloud layers are presented and compared between all measurement sites under study. First, the average profiles of aerosol optical and microphysical properties are presented. Then, instrumental detection thresholds, boundary layer effects and gravity wave activity are all analyzed as potential influencing factors on the retrieved ice formation characteristics.


### 3.1 Aerosol conditions at Leipzig, Limassol and Punta Arenas

To provide a general insight into the aerosol conditions at the three sites, the average optical and microphysical aerosol properties as derived from the lidar observations (Sec. 2.2) are shown in Fig. 3. The impact of aerosol on clouds is controlled more strongly by temperature than geometrical height, hence the averages are also calculated with temperature as a vertical coordinate.

### 3.1.1 Optical properties

The average aerosol backscatter coefficient $\beta_\mathrm{p}$ at $532\,\mathrm{nm}$ and the particle depolarization ratio derived from the Polly$^\mathrm{XT}$ observations with the Klett method (see Sec. 2.2) are investigated in this section. The evaluation of the air mass sources is covered more quantitatively in the next section and Appendix A.

The central European site of Leipzig is characterized by predominantly continental aerosol mixed with anthropogenic pollution (Baars et al., 2016). Long-range transport of dust may occur periodically, especially during spring and autumn (Ansmann
et al., 2003) as well as lofted smoke layers from wildfires (Haarig et al., 2018). Aerosol optical thickness (AOT) is derived from sun-photometer observations. Mean AOT at $500\,\mathrm{nm}$ at TROPOS, Leipzig, between 2018 and 2018 is $0.198$. When only the periods with co-located Polly$^\mathrm{XT}$ observations (used in this study) are considered, the AOT is $0.216$. Mean $\beta_\mathrm{p}$ at $532\,\mathrm{nm}$ drops below $0.2\,\mathrm{Mm^{-1}sr^{-1}}$ only above $4\,\mathrm{km}$ height, which corresponds to and extinction coefficient of $1.0\,\mathrm{Mm^{-1}}$ for continental aerosol conditions assuming a lidar ratio of $50\,\mathrm{sr}$.

Limassol is characterized by a distinct dry season with no precipitation and very few clouds during the summer. Generally, Limassol is frequently affected by aerosol transport from Africa, the Middle East and Europe with aerosol characteristics including dust (mineral and soil), marine (organics and sea salt) and anthropogenic pollution as well as mixtures of these (Nisantzi et al., 2015). Mean AOT at $500\,\mathrm{nm}$ is $0.176$ during the whole observational period and $0.165$ during the 'cloudy season' from October to May. In the following, the non-cloud season from June to September is excluded from the statistics.
The profile of mean backscatter is similar to the one at Leipzig within a factor of $1.5$, whereas the median generally is higher at Leipzig.

The aerosol load at Punta Arenas can be separated into two distinct layers with an aerosol-rich boundary layer and pristine conditions aloft. The free troposphere is dominated by marine aerosol from the Southern Ocean and was reported to show no changes compared to the pre-industrial conditions (Hamilton et al., 2014). Nevertheless, events of aerosol long-range transport
also occur occasionally (Foth et al., 2019; Floutsi et al., 2021). The boundary layer is laden with a mixture of marine and continental aerosol, as Punta Arenas is located $230\,\mathrm{km}$ inland from the Pacific coast. Mean AOT at $500\,\mathrm{nm}$ is $0.055$ during the whole campaign, but dropping to $0.047$, when excluding the period of long-range wildfire smoke transport in early 2020 (Ohneiser et al., 2020). Average boundary layer height is around $1.5\,\mathrm{km}$ (Foth et al., 2019) with negligible $\beta_\mathrm{p}$ above $2.0\,\mathrm{km}$ height ($90\%$ percentile of $\beta_\mathrm{p}$ at $532\,\mathrm{nm}$ dropping below $0.2\,\mathrm{Mm^{-1}sr^{-1}}$). Comparing the backscatter at Punta Arenas and
Limassol, the $90\%$ percentile at Punta Arenas is more than $30\%$ below the mean of Limassol and Leipzig at similar heights above ground. Both European sites show quite some variability, as well, with the $90\%$ percentile twice as large as the mean.





At temperatures above $5°C$, $\beta_p$ shows the strongest difference between the three sites (Fig. 3b), also explaining the larger AOT at Limassol and Leipzig. Between $0°C$ and $-12°C$, the mean $\beta_p$ at Limassol and Punta Arenas are almost equal. This counterintuitive behavior can be explained by different temperature regimes. The $-5°C$ isotherm at Punta Arenas varies be-

tween $1.1$ and $3.3\,km$ height, whereas at Limassol it varies between $3.0$ and $4.8\,km$ height and at Leipzig between $2.1$ and $4.8\,km$. At Punta Arenas, corresponding to the height of $2.0\,km$, only very low values of $\beta_p$ are observed at heights below a temperature of $-10°C$. The pronounced decrease of backscatter at Leipzig and Limassol is observed at slightly higher temperatures, which agrees with the, on average, higher boundary layer temperature there. Assuming typical lidar ratios of $50, 45, 20\,sr$ at Leipzig, Limassol and Punta Arenas, respectively, typical aerosol extinction coefficients can be estimated from the median

$\beta_p$ profile (Sec. 2.2). As shown in Fig. 3c, the extinction coefficients at Punta Arenas are a factor 2-3 lower than over Limassol for temperatures below $-10°C$. This difference decreases to a factor of 1.5 for slightly supercooled temperatures of above $-10°C$. Comparing Punta Arenas and Leipzig, the extinction is a factor of 3-4 higher at the latter site for all temperatures below freezing. This optical properties serve as a proxy for the background reservoir of aerosol particles that could act as cloud condensation nuclei and ice nucleating particles in the free troposphere.

The particle depolarization ratio provides hints on the aerosol particle types (Fig. 3d). At Leipzig, the depolarization ratio is approximately $0.05$ at all temperatures, typical for a continental aerosol with a slight contribution of mineral dust. The depolarization ratio at Limassol is bimodal, with one peak above $20°C$, a second one at $-14°C$ and a minimum at $4°C$. The first one can be ascribed to mineral dust in the boundary layer and originating from local sources. The second one is caused by long-range transport of mineral dust. At Punta Arenas, a maximum at $+8°C$ is caused by occasional events of dried sea salt

aerosol at the top of the atmospheric boundary layer (Haarig et al., 2017; Bohlmann et al., 2018). Going to lower temperatures, the depolarization ratio has a minimum of $0.01$ at $-20°C$ and a slight increase afterwards.

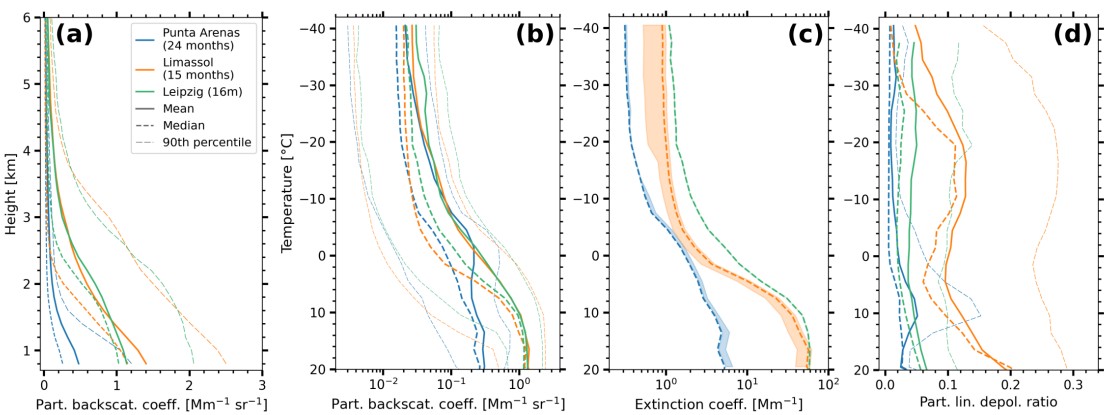

**Figure 3.** Profiles of average aerosol optical properties at $532\,nm$ wavelength derived by the PollyNET retrieval using the Klett method. Particle backscatter coefficient over height (a) and temperature (b) for Leipzig, Limassol and Punta Arenas. Extinction coefficient (c) and particle depolarization ratio (d) for the same locations. For the extinction coefficient only the median values based on the typical lidar ratio at each site are shown. Variability caused by different aerosol mixtures (see Sec. 3.1.2) is denoted by shading.



### 3.1.2 Contrasts in INP concentrations

An important aspect for the discussion of microphysical contrasts in clouds over the three sites is, how the differences in aerosol optical profiles are linked to contrasts in the INP load. An estimate of average INP concentrations covering the whole troposphere can be derived with the parametrizations described in Sec. 2.2. The decision for the aerosol typing required in the INP retrieval is based on the particle depolarization ratio and air mass source estimates. The air mass source methodology is based on Radenz et al. (2021) and shown in Appendix A.

At Punta Arenas marine sources are by far the most frequent, contributing $90\%$ to the residence time throughout the troposphere and peaking $95\%$ at $2.7\,\mathrm{km}$ height. Only below $2.0\,\mathrm{km}$ height (or above $-10°\mathrm{C}$) local terrestrial sources make up to $10\%$ of the air mass. Above $5\,\mathrm{km}$ height (below $-20°\mathrm{C}$) sparsely vegetated areas in Australia contribute up to $4\%$ to the air mass source. Mean particle depolarization ratios below $0.02$ in the free troposphere exclude frequent presence of mineral dust (Fig. 3d). Contrarily, at Limassol air masses with a marine source contribute $50$ to $80\%$ to the mixture. But these air masses are not pristine marine, as the eastern Mediterranean is enclosed by strongly populated landmasses. Below $5\,\mathrm{km}$ height (above $-16°\mathrm{C}$) continental Europe is second strongest source, responsible for up to $45\%$ at $0.5\,\mathrm{km}$ height. In the upper troposphere, barren ground from the Sahara is the strongest terrestrial source with contributions around $15\%$. POLIPHON (see Sec. 2.2) shows a peak at $-12°\mathrm{C}$ with mean dust fractions of $0.3$ ($90\%$ percentile $0.87$). Hence, the backscatter is divided into dust and non-dust according to the dust fraction. The non-dust portion is then split up into continental and marine, with the $40\%$ contribution of the continent above $-12°\mathrm{C}$ and $20\%$ below. The aerosol mixture at Leipzig is dominated by continental aerosol, with an average dust fraction of $0.1$.

The derived INP concentrations are depicted in Fig. 4. Note, shown is the INP concentration at a certain atmospheric temperature (also a proxy for height) based on the optical properties, aerosol types and the parametrization, not a freezing spectrum obtained from sampling a single air parcel and varying temperature.

At temperatures above $-10°\mathrm{C}$ average INP concentrations between $4\cdot10^{-3}$ to $6\cdot10^{-2}\,\mathrm{L}^{-1}$ can be expected at all three locations. With decreasing temperatures, the concentration increases to $0.1-1\,\mathrm{L}^{-1}$ at $-25°\mathrm{C}$ at the northern-hemispheric sites of Leipzig and Limassol. A strong increase of ice nucleating efficiency with decreasing temperature is counterbalanced by a decreasing aerosol concentration at lower temperatures, which are tied to greater heights. INP concentrations at Punta Arenas are strongly controlled by the fraction of continental aerosol in the free troposphere. Continental sources can contribute up to $1\%$ between $-11$ and $-24°\mathrm{C}$ and up to $4\%$ at lower temperatures. When only pristine marine aerosol is present, the INP concentration can expected to be 3-4 orders of magnitude lower than at the two other sites. As soon as very small fractions of continental aerosol are present, the INP concentration increases significantly. However, even if few percent of continental aerosol are assumed, throughout the heterogeneous freezing regime, INP concentrations remain a factor of 2-6 lower at Punta Arenas, compared to Leipzig and Limassol. Due to the absence of suitable remote-sensing or in-situ measurements, the actual contribution of continental aerosol to the free-tropospheric aerosol load over Punta Arenas can to date not be obtained. The range given in Fig. 4 is thus a solid estimate of the minimum and maximum of the expectable range of possible INP values. At temperatures above $-10°\mathrm{C}$, which refers to a height range that is frequently within the boundary layer, INP concentrations at




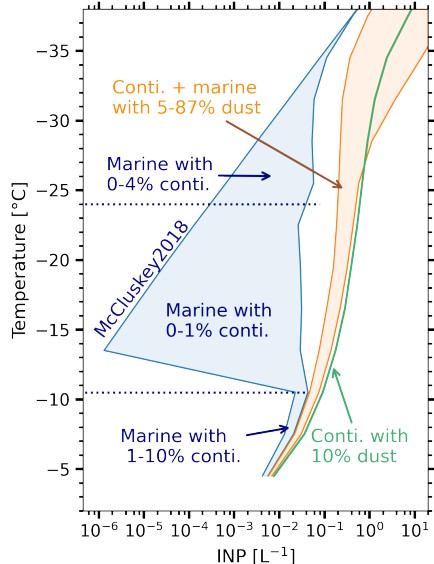

**Figure 4.** Average INP concentrations derived from optical properties for Leipzig (green), Limassol (orange) and Punta Arenas (blue). Shading shows the spread covered by different compositions. Temperature is used a a vertical coordinate. Fraction of continental is abbreviated 'conti.'.

Punta Arenas are high and within an order of magnitude to the concentrations at Leipzig and Limassol at the same temperature. Concentrations that are equally high as retrieved from the remote sensing observations above $-10\,°C$ were also found in situ at up-wind hilltop station on Cerro Mirador in an altitude of $622\,m$ above sea level (Gong et al., in preparation).

### 3.2 Mixed-phase stratiform cloud properties

The automated stratiform cloud selection algorithm introduced in Sec. 2.4 is applied to the Cloudnet datasets of all three locations. An overview over identified clouds and temperature-resolved phase occurrence frequency is provided in Sec. 3.2.1. Afterwards, it is shown how boundary layer aerosol load (Sec. 3.2.2) and gravity waves affect the derived phase occurrence (Sec. 3.2.3).

### 3.2.1 Phase occurence frequency and detection thresholds

Fig. 5 shows all the cloud objects that fulfill the criteria introduced in section 2.4. At all three locations, the fraction of profiles (i.e. fraction of time) in which a virga originated from the liquid layer increases with decreasing temperature. Similarly, the ice-to-liquid content ratio (Bühl et al., 2016) increases for lower temperatures. Liquid only clouds (fraction of profiles where no ice was produced from the liquid layer) were observed at Leipzig and Limassol down to $-16°C$. Whereas at Punta Arenas such clouds were observed even at temperatures as low as $-38°C$.



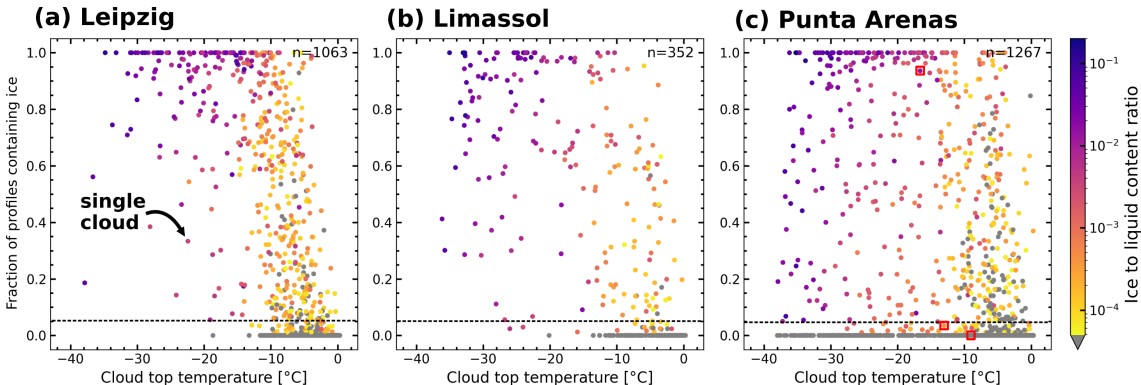

**Figure 5.** All stratiform clouds identified by the automated selection algorithm for Leipzig (a), Limassol (b), and Punta Arenas (c) by their cloud top temperature and the fraction of profiles, where ice is observed below the liquid-dominated cloud top. The $0.05$ threshold for the classification is marked by a dashed line. The color gives the median ice-to-liquid content ratio. $n$ gives the number of cloud objects in each dataset. Red rectangles in (c) denote the clouds also shown in Fig. 2.

The temperature-resolved fraction of occurrence of ice-forming clouds provide a first insight into the contrasts between different locations (Choi et al., 2010; Kanitz et al., 2011; Seifert et al., 2015; Tan et al., 2014; Zhang et al., 2018). For the phase occurrence frequency, a cloud is classified as ice-producing, if an ice virga was observed at least during $5\%$ of a cloud objects duration (Sec. 2.4).

Generally, more clouds contain ice more frequently for decreasing temperature (Fig. 6, solid lines). While at Leipzig and Limassol all clouds with CTTs below $-16°C$ contained ice, at Punta Arenas a fraction of $0.5-0.7$ of shallow stratiform clouds at these temperatures were classified as liquid only. This behavior is discussed further in Sec. 3.2.3.

Compared to prior lidar-based studies (e.g. Kanitz et al., 2011; Choi et al., 2010; Seifert et al., 2010), the fraction of ice containing clouds in the synergystic dataset is higher at temperatures above $-10°C$. At these temperatures, the amounts of ice produced and hence, the radar reflectivity and optical extinction are usually very low and stays undetected for lidar (Bühl et al., 2013a) and space-borne radars (Bühl et al., 2016). By utilizing established relationships between, radar reflectivity factor, temperature, IWC and optical extinction, such as the one from Hogan et al. (2006), one can estimate the sensitivity of a deployed lidar system to detect a certain mass of ice. In the case of the lidar data used in the Cloudnet classification, an extinction threshold of $12\,\mathrm{Mm}^{-1}$ had to be applied in order to best match the lidar-only statistics from Punta Arenas and Leipzig presented by Kanitz et al. (2011) (Fig. 6, dashed lines).

### 3.2.2 Effect of boundary layer aerosol load on phase occurence

As discussed in Sec. 3.1, the aerosol load at Punta Arenas is confined to the lowermost $2\,\mathrm{km}$ and the aerosol load at temperatures above $-10\,°C$ is similar to Limassol. To check for possible impact of this boundary layer aerosol on the ice formation efficiency, the basic temperature-resolved phase occurrence frequency (Fig. 6) is split into two, one containing cloud objects with bases below $2\,\mathrm{km}$ height and one with cloud objects having bases above that threshold (Fig. 7), in the following denoted as coupled





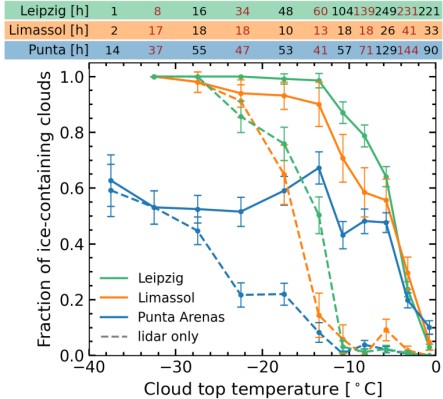

**Figure 6.** Fraction of ice containing clouds over temperature for Leipzig, Limassol and Punta Arenas. Total duration of the clouds in each bin is given by the numbers on top in hours. Dashed curves mark the occurrence frequency when using a lidar detection threshold of $12 \, \mathrm{Mm}^{-1}$.

and uncoupled clouds, respectively. At any height, temperatures vary by more than $11°$C (10 to $90\%$ percentile), providing ample coverage for the height threshold. Generally, coupled clouds show higher fractions of ice (absolute difference increases by $0.3$ at all locations at $-8°$C). The temperature-resolved phase occurrence frequency for the coupled clouds show rapid increase in fraction of ice-containing clouds, reaching $1.0$ at temperatures of only $-15°$C. The absolute difference between the fractions is less than $0.20$, with the lowest fractions still being observed at Punta Arenas. Below $-15°$C, almost no clouds

were observed at such low heights, especially at Limassol, hence no meaningful comparison can be done in this temperature interval.

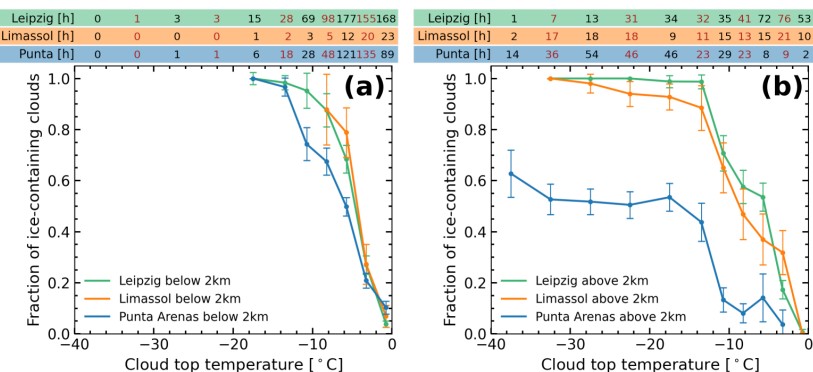

**Figure 7.** Fraction of ice containing clouds over temperature for Leipzig, Limassol and Punta Arenas, (a) below $2 \, \mathrm{km}$ and (b) above $2 \, \mathrm{km}$. Total duration of the clouds in each bin is given by the numbers on top in hours.

Considering only uncoupled clouds (cloud base above $2 \, \mathrm{km}$ height, 7b), stronger contrasts become evident. The fractions are similar at Leipzig and Limassol, but $0.15$ to $0.5$ at $-10°$C lower at Punta Arenas.



### 3.2.3 Gravity wave influence on phase occurence at low temperatures

The lack of ice-containing cloud layers at temperatures below $-18°$C over Punta Arenas is a prominent feature of both, the lidar-radar and the lidar-only-equivalent datasets shown in Fig. 6. The associated excess of supercooled liquid water is frequently reported as a general phenomenon of stratiform clouds in the southern hemisphere mid and higher latitudes. Within this subsection, the reasons for the found behavior over Punta Arenas will be elaborated in more detail.

Frequently, the stratiform liquid-only cloud layers observed over Punta Arenas at temperatures below $-18°$C are embedded
in orographic gravity waves. Following the wave detection methodology introduced in Sec. 2.5, Fig. 8 shows the autocorrelation and power spectra of the Doppler lidar vertical velocity for clouds classified as liquid only over Leipzig, Limassol and Punta Arenas, respectively. These supercooled liquid-only clouds at Punta Arenas show high autocorrelation coefficients at long shifts (Fig. 8c1), whereas liquid only clouds with similar characteristics are absent at Limassol (Fig. 8b1) and Leipzig (Fig. 8a1). In terms of spectral power density (Fig. 8c2), the strongly supercooled clouds at Punta Arenas show only low turbulence.

By decreasing the length threshold at which the autocorrelation drops below $0.8$, weakly wave-influenced clouds are gradually removed from the phase occurrence frequency statistics. Fig. 9 shows an increase in fraction of ice-containing clouds below $-12°$C with decreasing correlation length thresholds from $30000$ to $300$ m. At Punta Arenas, the fraction increases from $0.5$ to $0.85$. In the temperature interval between $-15$ and $-25°$C the fraction is $0.05$ to $0.1$ lower at Punta Arenas compared to Leipzig and Limassol. Hence, clouds with fully developed turbulence show similar ice formation frequencies, independently
of the location, with indications for a still slightly reduced ice formation efficiency over Punta Arenas.

### 3.2.4 Comparison of radar reflectivity factor of the ice virga

Prior studies of Zhang et al. (2018) identified a strong contrast in radar reflectivity factor between the different 30-deg latitude bands of the globe, with the southern hemisphere mid-latitudes (30-60°S) showing the lowest mean reflectivity of all regions. They concluded that this difference in reflectivity factor is associated to a respective difference in ice crystal mass and number
concentration. In the following we provide a similar representation of regional contrasts of ice-virga reflectivity from ground-based perspective that is based on a single radar instrument. As described in Sec. 2.4, the amount of ice formed in the mixed phase layer is measured at six height bins ($180$ m) below the base of the liquid dominated cloud top and hence at the top of the virga (Bühl et al., 2016). Fig. 10a shows the cloud top temperature-resolved statistics of reflectivity for the three stations, which is based on the full cloud dataset, including the wave-influenced clouds (as these are also included in the study by Zhang
et al. (2018)). From $-28°$C to $-16°$C and again above $-12°$C, Punta Arenas shows the lowest reflectivity and Limassol the highest. For most temperatures, Punta Arenas is $5-8$ dB below the northern hemispheric stations, only exception is the interval between $-4$ and $-8°$C, where the reflectivity for Punta Arenas and Leipzig is almost equal. At temperatures below $-28°$C, Limassol and Punta Arenas show equal reflectivity with values at Leipzig being slightly higher.

We further investigated the properties of the ice-forming liquid-dominated cloud top layers and found that cloud thickness
agrees within $40$ m above $-30°$C. The ice-to-liquid content ratio (Fig. 10b) is smaller at Punta Arenas, than at the northern



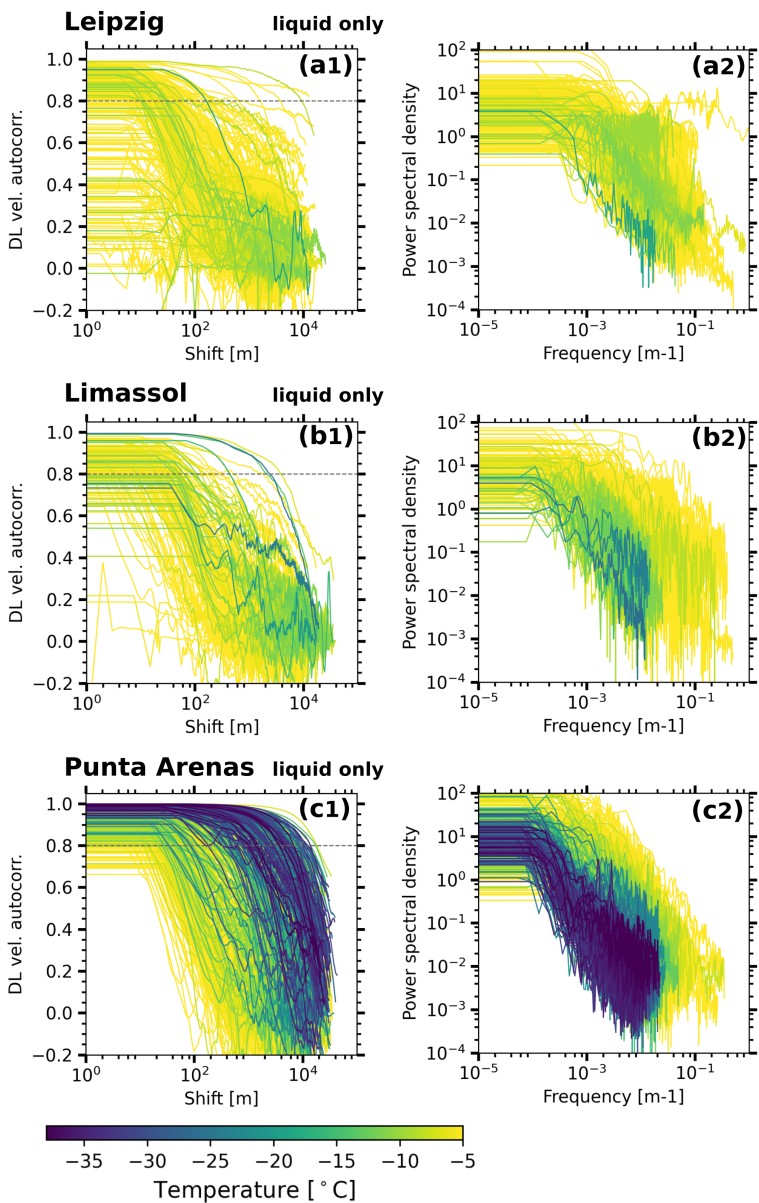

**Figure 8.** Autocorrelation function (a1, b1, c1) and power density (a2, b2, c2) of the Doppler lidar vertical velocities for cloud objects classified as liquid only. The columns are Leipzig (a), Limassol (b) and Punta Arenas (c). Color gives cloud top temperature.

hemispheric locations, especially (factor 3) between $-24$ and $-20°C$, but also above $-10°C$. Hence, in these temperature regimes, the liquid phase is less efficiently converted into solid.





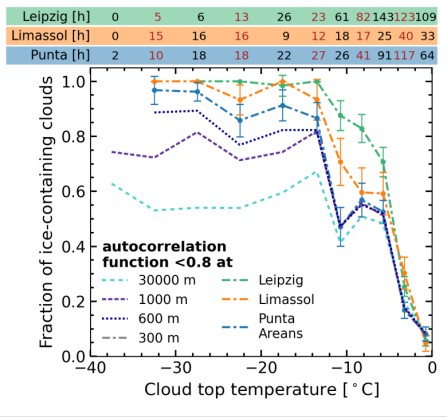

**Figure 9.** Fraction of ice containing clouds over temperature for Leipzig, Limassol and Punta Arenas. Dash-dotted lines show the fractions for Leipzig, Limassol and Punta Arenas with an autocorrelation coefficient smaller than $0.8$ for a horizontal shift of $300\,\mathrm{m}$. The fractions for cloud objects with longer autocorrelation are only shown for Punta Arenas. Total duration of the clouds in each bin is given by the numbers on top in hours.

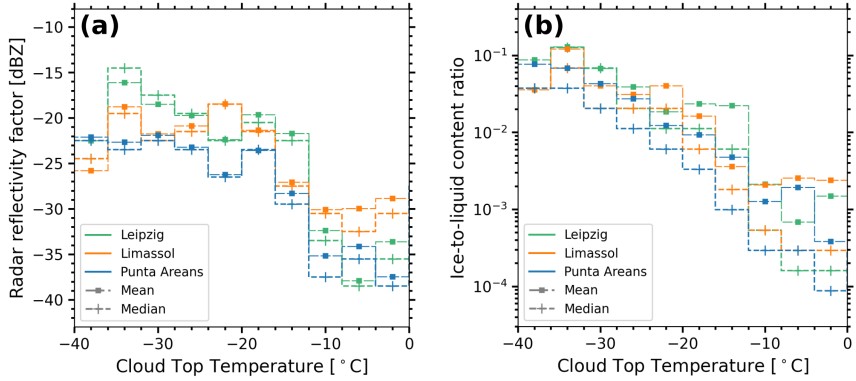

**Figure 10.** Radar reflectivity factor observed at the top of the virga ($180\,\mathrm{m}$ below the liquid-dominated layer base) (a) and Ice-to-liquid-content-ratio (b) binned by CTT. Temperature bins of $4^\circ\mathrm{C}$.

## 4  Discussion of observed contrasts in properties of shallow clouds

In the previous section, a comprehensive overview about the aerosol conditions and stratiform cloud properties at the strongly

contrasting sites of Leipzig, Limassol and Punta Arenas was presented. Several aerosol-, temperature-, surface-, as well as dynamics-related differences have been identified and will be discussed in the following.

The analysis of the profiles of aerosol optical properties obtained from the Polly$^\mathrm{XT}$ lidar observations revealed almost similar aerosol load at Punta Arenas and Limassol between $0$ and $-10^\circ\mathrm{C}$. Leipzig shows a higher aerosol load in this temperature





interval. For lower temperatures the lowest average extinction was observed at Punta Arenas. Low values of particle depolar-
ization ratio reveal, that non-spherical particles, such as mineral dust are completely absent at Punta Arenas below $-10°$C. The
absence of aerosol species in the free troposphere is also consistent with the Southern Ocean south of Australia (e.g. Alexander
and Protat, 2019). Together with an air mass source estimate, the average profiles of optical parameters were used as an input
for INP parametrizations for a first estimate of long-term averages of INP concentrations. The lidar-based estimate of average
INP concentrations identifies the strongest differences in INP concentrations between $-12$ and $-35°$C. However, it is still
subject to discussion which fraction of marine particles is transported from the boundary layer to the free troposphere. This
insufficient knowledge propagates further into uncertainties in the fraction of continental aerosol, which leads to considerable
uncertainties in the estimated concentrations of INP. Bourgeois et al. (2018) argument, that due to efficient removal processes,
marine particles predominantly remain in the boundary layer (globally 80% of their AOT). Murphy et al. (2019) argument in a
similar direction, especially when sea salt is regarded a tracer of marine origin.

Opposed to prior lidar-only studies (e.g. Kanitz et al., 2011; Alexander and Protat, 2018), more than half of the clouds
observed at $-8°$C at the three sites were found to contain ice. This is in line with the lidar detection threshold introduced by
Bühl et al. (2013a), but extending their results to Limassol and Punta Arenas. This result has significance for lidar-only or
ceilometer datasets that are currently used for validation of climate models (e.g. Kuma et al., 2020). The occurrence of ice at
these relatively warm temperatures is also in line with airborne in-situ observations over the Southern Ocean (Huang et al.,
2017; D'Alessandro et al., 2019).

Ice formation at slightly supercooled temperatures above $-10°$C was found to be equally frequent, suggesting the pres-
ence of equally efficient INP at all sites. At Punta Arenas, such temperatures usually occur within the boundary layer, where
presumably INP of biological origin are present (Gong et al., in preparation). Similarly high fractions were also observed for
Arctic boundary layer clouds during summer and autumn (Achtert et al., 2020). Griesche et al. (2020) also found, that surface
coupling increased the frequency of ice formation in boundary layer clouds observed during Arctic summer.

The liquid-only clouds at Punta Arenas found below $-14°$C are identified to be predominantly mountain wave clouds,
likely in an early period of their lifecycle. In this kind of cloud, the liquid-dominated mixed-phase is separated horizontally
from the ice virga, due to the rapid flow and weak turbulence. When sampled by stationary observations, only parts of the
clouds are covered and the phase occurrence frequency is biased. Using Doppler lidar vertical velocity observations, clouds
driven by orographic wave dynamics can be identified based on their autocorrelation function, as presented in Sec. 3.2.3.
At temperatures below $-14°$C, approximately 1/3 of the cloud layers observed at Punta Arenas were influenced by gravity
waves. Similar frequencies of occurrence of non-turbulent liquid layers were also found at Utqiaġvik (Alaska) and McMurdo
(Antarctica) deriving stability criteria form radiosoundings (Silber et al., 2020).

In a final step, the separation techniques for coupling and orographic waves can be combined, to assess contrasts ice fre-
quency for free-tropospheric and fully turbulent clouds. The resulting occurrence frequency is shown in Fig. 11. The fraction
of ice containing clouds at temperatures below $-15°$C is above $0.85$ at all three sites. However, comparing to Leipzig and Li-
massol, the fraction of ice forming clouds with CTT between $-25$ and $-15°$C remains $0.1$ lower at Punta Arenas. Mineral dust
is known to be an efficient INP at these temperatures (Kanji et al., 2017), but was not observed at the respective temperatures at





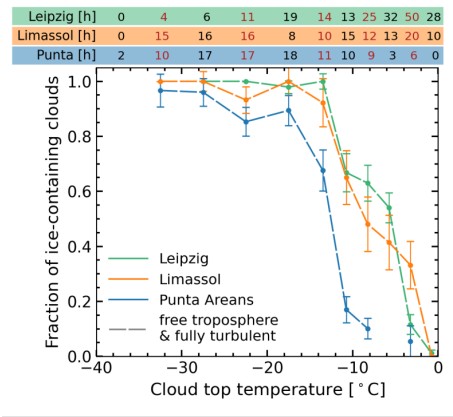

**Figure 11.** Fraction of ice containing clouds over temperature for Leipzig, Limassol and Punta Arenas when considering only fully turbulent clouds with an autocorrelation coefficient smaller than 0.8 for a horizontal shift of 300 m (see Fig. 7) and cloud bases in the free troposphere above 2 km height (see Fig. 9).

Punta Arenas (Fig. 3). This difference in the non-wave cloud phase occurrence is also in agreement with satellite-based studies

of Villanueva et al. (2020), who also attributed latitudinal differences in the ice occurrence to associated differences in the dust load. Fig. 11 also depicts the low frequency of ice formation in free-tropospheric fully turbulent clouds at temperatures above $-15\,°C$. With less coupling to the near-surface aerosol reservoir, ice formation is strongly suppressed compared to Leipzig and Limassol.

The ice mass formed by stratiform liquid layers is lowest at Punta Arenas, as was found by investigating the radar reflectivity

factor in the ice virga as a proxy. Especially between $-28°C$ and $-16°C$, the observed radar reflectivity factor is up to 7 dB lower, compared to both sites in the northern hemisphere. This result is consistent with estimates from space-borne sensors covering the full Southern Ocean (Zhang et al., 2018). Contradicting, Arctic mixed-phase clouds were found to respond with lower IWC to increased loads of anthropogenic pollution (Norgren et al., 2018). As for the lower frequency of ice formation discussed above, the difference in ice mass coincides with a lack of dust INPs at these temperatures. Slight differences in

IWC or Z, respectively, might be explained by a slower mass growth rate, caused by a smaller vapor diffusion coefficient at higher ambient pressure (Hall and Pruppacher, 1976). When temperature and particle size are considered similar, the stratiform clouds subject to this study, experience 10 to 20% larger growth rates at Leipzig and Limassol than at Punta Arenas. Hence, the difference of a factor 3-6 larger ice mass cannot be explained solely by this effect alone.

Frequent occurrences of ice forming clouds above $-10°C$ were found, which were not covered by studies based on space-

borne active remote sensing. Similar to their colder counterparts, they show a lower ice amount in the virga above Punta Arenas, but with smaller difference compared to the sites in the northern hemisphere. With an average reflectivity of $-36\,dBZ$, they are usually well below the detection limit of the CLOUDSAT satellite in the A-Train constellation (Bühl et al., 2013a, 2016). Occurrence of such clouds in other parts of the Southern Ocean cannot be ruled out. A misclassification of supercooled drizzle





clouds as ice containing is unlikely, as they exceed $-30\,\mathrm{dBZ}$ neither at cloud top, nor in the virga. From previous studies it is
known that the onset of drizzle formation is usually associated to higher reflectivities, either above approximately $-20\,\mathrm{dBZ}$
(Liu et al., 2008; Acquistapace et al., 2019) or at least above $-30\,\mathrm{dBZ}$ (Wu et al., 2020).

## 5  Summary and Outlook

This study investigated contrasts in aerosol-cloud interactions in shallow supercooled stratiform clouds observed with the
ground-based remote sensing supersite LACROS at Leipzig, Limassol and Punta Arenas.

Sampling the profiles of optical properties with temperature as a vertical coordinate revealed aerosol load at temperatures
between $-15$ and $0\,°\mathrm{C}$ being within a factor of 2 at Punta Arenas and Limassol. This finding is related to the cold and (compared
to the free troposphere) aerosol laden boundary layer at Punta Arenas. At lower temperatures, the lowest $\beta_\mathrm{p}$ and extinction was
observed over Punta Arenas, the highest over Leipzig. The very low particle depolarization ratio at Punta Arenas between $-25$
and $-10\,°\mathrm{C}$, suggests the absence of mineral dust in a temperature regime, where dust is known to be an efficient INP. An
estimate of INP concentrations at the respective temperatures based on the optical properties reveals differences of 1-4 orders
of magnitude between Punta Arenas and the two northern hemispheric sites. In absence of abundant INPs from marine sources,
the contribution of terrestrial sources causes strong variability.

The phase occurrence frequencies showed a higher fraction of ice containing clouds at weakly-supercooling temperatures
of above $-10\,°\mathrm{C}$ compared to prior lidar-only studies. A cloud radar with a sensitivity better than $-40\,\mathrm{dBZ}$ is needed to
sufficiently characterize low ice water contents in the virga formed by shallow stratiform clouds in this temperature regime.
Coupling to the boundary layer increases frequency of ice formation at slightly supercooled temperatures at all sites. The
strongest contrasts in the ice formation frequency between free-tropospheric and surface-coupled conditions were found for
Punta Arenas. This finding is in compliance to the found contrasts in the INP profiles at the three sites and further indicates
that the free-tropospheric INP reservoir over the Southern Ocean is limited.

Frequent liquid only layers below $-20\,°\mathrm{C}$ at Punta Arenas were found to be associated with gravity waves, causing two
implications: (1) potential phase misclassification by stationary observers due to horizontal separation of ice and liquid phase;
(2) sustained liquid water in updrafts, because the associated vertical velocities allow supersaturation over water; The newly
developed Doppler lidar autocorrelation approach helps to address "[...] the difficulty in characterizing gravity waves from
existing single-site measurements [..]" as posed by Silber et al. (2020). Ice mass in the virga and the ice-to-liquid content
ratio was found to be lowest at Punta Arenas, especially at temperatures where dust serves as an efficient INP at the northern
hemispheric sites.

From the results of this study, two issues for further investigations arise. Firstly, importance of terrestrial sources of INP
in the Southern Ocean. Observations of free-tropospheric aerosols upwind and downwind of major landmasses are required,
to quantify the terrestrial emission and identify regions, where the emitted INP impact cloud microphysics the strongest.
Secondly, how important are gravity waves in the formation of clouds in other regions of the Southern Ocean? How frequent
do gravity waves occur above the open ocean, which covers the vast majority of the Earths surface between 30 and $70\,°\mathrm{S}$? Such





measurements are required in order to further constrain the role of INP in the evident excess of liquid water in clouds of the southern hemisphere mid-latitudes.

*Code and data availability.* The Cloudnet datasets are provided by the ACTRIS Data Centre node for cloud profiling under the following

links: https://hdl.handle.net/21.12132/2.e130b236928a4932 (Leipzig), https://hdl.handle.net/21.12132/2.a056a828a6b94d1f (Limassol) and https://hdl.handle.net/21.12132/2.b6c194d7d33b448e (Punta Arenas). In the upcoming future, the Doppler lidar and PollyNET datasets will also be available via ACTRIS. Meanwhile, they can be obtained upon request at (polly@tropos.de). The source code of the cloud sniffer, the database of identified clouds and analysis scripts are available at https://github.com/martin-rdz/larda_cloud_sniffer (last accessed 27.04.2021; Radenz and Bühl, 2021) and requires a setup of pyLARDA (https://github.com/lacros-tropos/larda, last accessed 27.04.2021;

Bühl et al., 2021). AERONET photometer observations at Leipzig, Limassol and Punta Arenas are available from the AERONET database (http://aeronet.gsfc.nasa.gov/). We thank AERONET-Europe for providing the calibration service.

## Appendix A: Air mass source estimate

The automated time–height-resolved air mass source attribution described by Radenz et al. (2021) is used to characterize air mass origin for the LACROS observations. As in the original publication, 10-day HYSPLIT ensemble backward trajectories

are calculated in intervals of 3 hours and $500\,\mathrm{m}$ throughout the period of the deployment. For the calculation of the residence times, a reception height of $2\,\mathrm{km}$ is used. Additionally to the MODIS based land cover classification and the the named geography (individually for Leipzig, Limassol, and Punta Arenas), $30°$-wide bands of latitude are used to characterize meridional transport. The assignment of the land cover classed and named geography are depicted in Fig. 2 and 3 of Radenz et al. (2021). The average residence time for each set of surface types is shown in Fig. A1. However, as for the aerosol optical properties, ge-

ometrical height is less insightful as temperature. Thus, the residence time at each $3\,\mathrm{h}$ profile is binned by ambient temperature, using analysis profiles. Then, the average residence time is calculated per temperature bin (Fig. A2).

*Author contributions.* MR analyzed the data and drafted the manuscript. MR, PS, JB, HB, RE, BBG, REM, and FZ conducted the campaigns and operated LACROS. PS, MR, JB generated the Cloudnet dataset. HB processed the Polly$^{\mathrm{XT}}$ lidar data. PS, JB, and AA supervised the work and revised the manuscript. All authors jointly contributed to the paper and the scientific discussion.

*Competing interests.* No competing interests are present.

*Acknowledgements.* The authors wish to thank TROPOS, Cyprus University of Technology and University of Magallanes for their logistic and infrastructural support during the LACROS deployments. We furthermore thank Teresa Vogl, Willi Schimmel, Audrey Teisseire, Cristofer





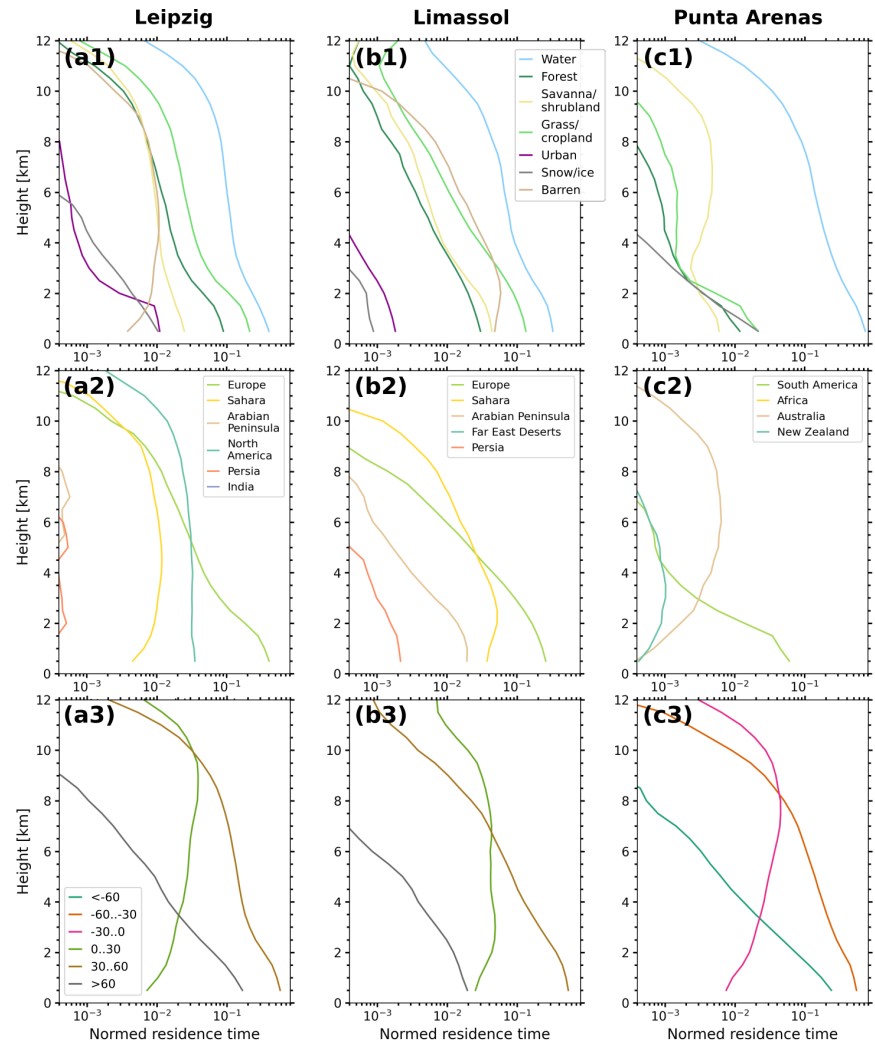

**Figure A1.** Mean normalized residence time profile at Leipzig (a), Limassol (b) and Punta Arenas (c) for a reception height of 2 km. Residence times are shown for the land cover classification (1), named areas (2) and latitude bands (3).

Jimenez, Heike Kalesse, and Roland Schrödner for keeping LACROS operational at Punta Arenas. We gratefully acknowledge the ACTRIS Cloud Remote Sensing Unit for making the Cloudnet datasets publicly available. LACROS operations were supported by the European Union (EU) Horizon 2020 (ACTRIS; grant no. 654109) and the Seventh Framework Programme (BACCHUS; grant no. 603445). Observations at Leipzig were supported by the BMBF funded projects "High Definition Clouds and Precipitation for Climate Prediction – HD(CP)2" (grant nos. 01LK1503F,01LK1502I, 01LK1209C, and 01LK1212C). Part of the CyCARE and DACAPO-PESO campaigns were funded by the Deutsche Forschungsgemeinschaft (DFG – German Research Foundation) project PICNICC (SE2464/1-1 and KA4162/2-1). JB acknowledges funding by DFG project COARSEMIX (398285025). REM has been financial supported by the SIROCCO project (grant no. EXCELLENCE/1216/0217) co-funded by the Republic of Cyprus and the structural funds of the European Union for Cyprus through



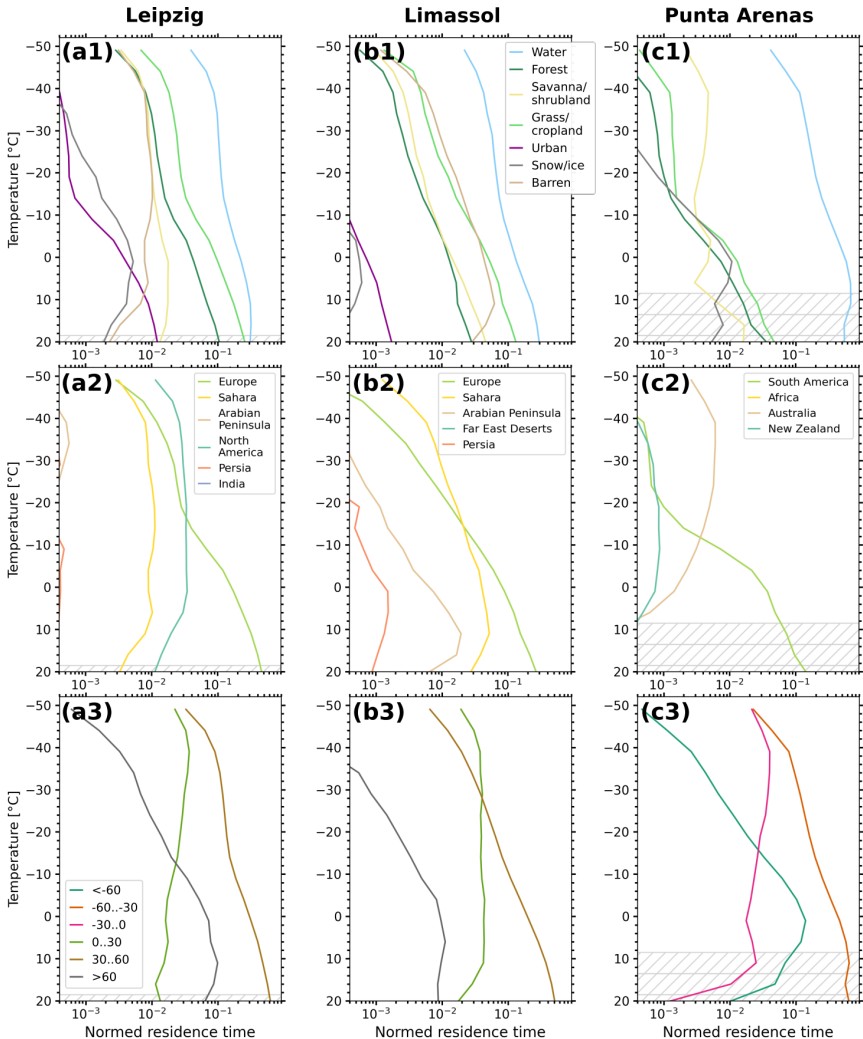

**Figure A2.** Mean normalized residence time binned by temperature at Leipzig (a), Limassol (b) and Punta Arenas (c) for a reception height of 2 km. Residence times are shown for the land cover classification (1), named areas (2) and latitude bands (3). Hatching indicates bins with less than 1 % of the values.

the Research and Innovation Foundation. Thanks are also given to EXCELSIOR EU H2020 Widespread Teaming program with Grant Agreement No 857510 (www.excelsior2020.eu) and the Republic of Cyprus for the support of the establishment the ERATOSTHENES Centre of Excellence. Thanks are also given to EXCELSIOR EU H2020 Widespread Teaming program with Grant Agreement No 857510 (www.excelsior2020.eu) and the Republic of Cyprus for the support of the establishment the ERATOSTHENES Centre of Excellence. BBG acknowledges funding by the ANID/CONICYT/FONDECYT Iniciación 11181335. HB acknowledges funding by the AEOLUS Cal/Val activities (BMWi; grant no. 50EE1721C).





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
