# Peer review of "Hemispheric contrasts in ice formation in stratiform mixed-phase clouds: Disentangling the role of aerosol and dynamics with ground-based remote sensing"

_Atmospheric Chemistry and Physics, 2021_

## Author Comment (AC1)

**General Remarks**

We thank both reviewers for their time and the constructive comments, which improve the quality of our manuscript significantly. The referee comments are formatted in grey and our reply in black with indentation and numbering (R##). The line and figure numbers refer to the revised version.

**Specific Reply to Referee #1**

This manuscript presents the role of aerosol and dynamics on ice formation in stratiform mixed-phase clouds based on the comparison of ground-based remote-sensing stations (one in the South hemisphere and two in the North hemisphere). The topic is well introduced. Despite the methodology is complex, the authors have made a good work, being quite easy to follow. Results and discussion sections are also well structured, helping to identify the main findings. Additionally, the paper is written in good English and, from my point of view, this paper makes an excellent contribution to the research field of aerosol-cloud interaction. Therefore, I suggest the acceptation of this manuscript with minor revisions.

Minor comments:

The title seems to attribute the contrasts in ice formation to the hemispheric location of the stations. Punta Arenas (South) vs Limassol and Leipzig (North). However, I would say that the contrasts can be finally attributed to 1) pristine vs 'dirty' atmospheric conditions and the gravity wave occurrences. I mean, a similar station (as Punta Arenas) but located in the North hemisphere would not show similar results?

> **R1:** That's an interesting point, raised by Referee #1. Indeed, one of the next steps would be to locate a northern-hemispheric mid-latitude site, which shows similar dynamic conditions as Punta Arenas while being exposed to the high northern-hemispheric aerosol concentrations. Perhaps the lee of the Scandinavian mountains (Scotland, Sweden, Finland) would be an option. Or some place in north-western parts of North America. Similarly, we are indeed already preparing for a remote-sensing campaign in southern New Zealand, were we expect stronger perturbations of the aerosol load caused by emissions of the Australian continent, while orographic effects are expected to be similar.

P4 L114-L117: The phrase 'Afterwards, the methods for cloud selection and vertical velocity characterization are introduced.' Seems to be redundant.

> **R2:** We corrected the text at the respective position.

P4 L120 (suggestion): Boundary layer coupling and vertical dynamics are discussed in Sec. 3.2.2 and Sec. 3.2.3, respectively.

> **R3:** Included as suggested.

P7 L169: Cloud Top Temperature is crucial in this paper. However, it is obtained from models (ECMWF's IFS analysis and GDAS) instead of from the microwave radiometer. Why? Which is the uncertainty of the Cloud Top Temperature? Since findings (chapter 5) are formulated at -5, -10, -15, -20ºC, the uncertainty should be below 5ºC

> **R4:** Similar to our previous studies (see, e.g., Seifert et al., 2010, Sec. 2.3; Kanitz et al., 2011, Sec. 2), we checked for the accuracy of the model temperatures by comparing the model values to available co-located radiosondes. In agreement to what was found previously, the uncertainties in the free troposphere (2-10 km) for Punta Arenas and Limassol are in the order of 1 K. Higher deviations can be expected in the boundary layer and in the tropopause

region. Nevertheless, as also suggested by Referee #2, we checked for the impact of the temperature uncertainty on the frequency of ice-formation statistics with a Monte Carlo sampling approach. A random error drawn from a uniform distribution between -2 and 2 K is added to the CTT. The resulting statistics are shown in Fig. 1 below. The findings described in the manuscript also hold for a 4 K uncertainty in the temperature.

[Figure]

**Fig 1:** Impact of random errors in CTT on fraction of ice containing clouds at Leipzig, Limassol, and Punta Arenas similarly to the figures shown in the manuscript. (a) basic statistics and lidar-extinction threshold, (b) filtered for wave clouds with the autocorrelation threshold, (c) cloud base below 2 km, (d) cloud base above 2 km, and (e) for free tropospheric and fully turbulent clouds.

**Specific Reply to Referee #2**

Summary:

Radenz et al. use a comprehensive suite of observations for three contrasting ground-based locations in order to quantify and understand differences in ice formation. They conduct a comprehensive analysis of aerosol and cloud properties derived from the ground-based sensors in order to reach four main conclusions: (1) average backscatter is fairly close between sites; (2) ice formation at relatively warm temperatures is more common than previously thought; (3) clouds coupled to the aerosol-rich BL show more ice formation; (4) gravity waves form liquid clouds over Punta Arenas and bias the cold statistics. I commend the authors for their thorough analysis and note that the four key points I summarised above (From their abstract and conclusions) are all important conclusions.

Although I suggest 'major revision' due to one major comment directly below, I think that this can easily be implemented (along with the minor comments).

I would strongly support acceptance of the revised manuscript by ACP. It will be a very good contribution to the literature.

Major comment:

Line 263. On the use of the Fast Fourier Transform (FFT). The authors described how the vertical velocity time-series is the input into the FFT in order to calculate power spectral density. However, in lines 253-254, the authors state that 'vertical velocity is sampled at the pixel with the maximum backscatter out of the heights…', and illustrate this as the dashed line in Figure 2c. Note that this varies in height. In other words, the authors are mixing a height and time series as use for input for the FFT. The series for input into the FFT should be time- only: it is not correct to incorporate information from a range of altitudes in your time-series. While I suspect these effects are likely small and won't change the conclusions, the authors should reprocess their data, using the vertical velocity at a constant altitude, for each cloud event. I leave it to Radenz et al. to choose what the altitude is for each cloud, but if you would like a suggestion, then perhaps use the mean (or median) altitude of peak backscatter for each cloud. Please propagate these changes through the manuscript.

> **R5:** Thanks for the suggestion. We originally implemented the height-adaptive sampling at the maximum of the backscatter signal to avoid gaps in the timeseries. We revised the scheme following your requestand now sample the middle of the liquid-dominated cloud top layer. The description of the methodology and Fig 2c is revised accordingly. Indeed, the results shown in Fig. 9 did not change significantly.

Minor comments:

Line 83: McFarquhar et al. paper is now published (2021), see https://doi.org/10.1175/BAMS-D-20-0132.1

> **R6:** Replaced as suggested.

Table 2: I struggle to believe that Limassol's climate is 'northern tropics'. For a start, it's at 35N, well outside the tropical region. I suggest you change this phrase to more accurately reflect the climate zone Limassol is in.

> **R7:** We changed it to Northern subtropics, which reflects the climate zone better.

Line 190: You use ECMWF IFS analysis for temperatures over Punta Arenas. The lack of radiosondes is somewhat unfortunate, but I fully agree with the authors in using an analysis product instead. So, some comment on the uncertainty in temperatures at cloud top height should be included though. While I doubt these uncertainties would often push your results into adjacent 5C-wide temperature bins, the

IFS temperatures will carry some uncertainties. Perhaps a Monte-Carlo simulation could be performed to test how a 1C or 2C uncertainty in cloud top temperature, of your thin stratiform clouds, changes the results. From my experience in comparing remote-sensing retrievals (radar and/or lidar), the (re)-analysis temperatures do not always match up with the melting level or cloud top inversion height.

**R8:** Thanks for sharing your experience. Referee #1 raised a related issue, please see R4.

Line 255 onwards: Can you add a figure, perhaps in an appendix, showing a typical example of clouds which are influenced by orographic waves please? It would be informative to visually see the differences as observed by your ground-based instruments.

**R9:** Thanks for the suggestion; we added an example case in the appendix (new Fig. A1).

Line 267: You need to explain why you select the 0.8 autocorrelation threshold. What happens when you vary this by e.g. changing to 0.7 or 0.9? How do the results change?

**R10:** The value of 0.8 is based on visual inspection of the whole dataset. When, changing the value at which the characteristic drop is estimated, the general results would not change, only the length of the shifts at which the autocorrelation threshold drops below the threshold value will change. For example, a value of 0.7 would require slightly larger shifts than 300 m to obtain a similar fraction of ice containing clouds. Defining a physically justified threshold would require a full characterization of the turbulent flow around the cloud. We included a brief discussion of this issue in the final paragraph of section 2.5. As also stated in response R11, we are working on a successor study about the properties of orographic waves.

Line 268 (approx.): The autocorrelation method is a good idea to determine the influence of orographic waves on cloud phase, and, as the authors describe, necessary at locations like Punta Arenas. To provide the broader context for wave forcing of clouds over Punta Arenas, and an additional verification on their method, I suggest that the authors composite the synoptic scale meteorology. I'd suggest using a reanalysis (ERA5?) at the closest time-step to the observed clouds, and compositing surface pressure and 10 m wind fields (as vectors) for the two cases of orographically-forced waves (autocorrelation>0.8, using your threshold in the paper) and no forcing (<0.8). In general, orographic waves form when relatively strong near-surface winds impinge upon a mountain range approximately perpendicularly (within 45degrees of this, e.g. see Section 5 in Dornbrack et al., 2001, JGR, https://doi.org/10.1029/2000JD900194 ). This is commonly observed in the Andes further north which are aligned north-south, but, around Punta Arenas, as the authors know, the topography is very complicated. The southern tip of the Andes is likely a unique region for cloud phase in the whole Southern mid-latitudes, so I think that understanding some of the synoptic features which generate the waves and result in enhanced amounts of liquid water would be useful for the community.

**R11:** Thank you for posing this interesting idea. Also from our current understanding, the occurrence of wave clouds is connected to specific meteorological conditions. But we have the feeling that incorporating the analysis of another dataset is beyond the scope of this study. We will definitely take a look at this topic in a follow-up study.

Figure 8: This is a nice, and informative, figure. However I wonder whether it's possible to increase readability, without losing the message, by plotting the average autocorrelations and PSDs in e.g. five degree temperature bands?

**R12:** Thank you for this comment. The curves of single clouds are now averaged into temperature bins of 5K.

Line 354: You could cite McFarquhar et al. (2021) here to support this point, who showed the measured INP values over the Southern Ocean and who also noted the ~3 orders of magnitude lower values than were reported in the 1970s.

**R13:** Done as suggested.

Line 402-403: You could expand your comments on Figure 7b as you only have 1 sentence at the moment.

**R14:** Thank you for the suggestion. We expanded the discussion of Fig. 7b.

Line 464: To support the statement that INPs in the BL are of biological origin over the Southern Ocean, how about citing Uetake et al. (2020) https://doi.org/10.1073/pnas.2000134117

**R15:** Added the reference. Thanks for the suggestion.

Technical corrections:

**R16:** Thank you for your technical suggestions. We implemented all of them as recommended.

Line 70: '…higher aerosol load allows us to advance…'

Line 84: 'But, apart from the year-long…'

Line 103: 'The goal of this…'

Line 113: '…and allow us to…'

Figure 2: Suggest you mention in the caption where the temperature is obtained from to avoid the reader having to dig through the text. You should also describe in the caption what the dashed line in Figure 2c is.

Line 241: '… horizontal extent of…'

Figure 5: Caption: 'red squares in (c)…'

Line 380: 'Limassol nearly all clouds…'

Line 380: Why not say '…fraction of (0.6 +/- 0.1) of shallow…'

Line 405: Well, 'The lack of ice-containing cloud layers…' seems pretty strong statement to me. Looking at Figure 7b, you still have them over Punta Arenas for about half of the clouds. Suggest rephrasing to 'The reduced fraction of ice-containing clouds…'

Line 446: 'absence of continental aerosol species…'

Line 453: 'Murphy et al. (2019) argue in a …'

Line 497: 'CloudSat'

Line 520: '… with orographic gravity waves…'

Figure A1 & A2: Add units to x-axes

**R17:** The unit of the normalized residence time is [s s$^{-1}$]. Thus we omitted it.

**Further changes**

A few changes were incorporated due to additional feedback.

1. The discussion of the Cotton and Field 2002 paper was misleading. We clarified that sentence (L244f original draft).
2. The caption of Tab. 3 was slightly expanded and the ice extinction, later used for the lidar detection threshold, was added.
3. The instrument naming in Tab. 1 was slightly refined.
4. Replaced 'cloud object' with 'cloud case' and 'land cover' with 'surface cover' throughout the manuscript.
5. We refined a few formulations throughout the manuscript, to make them clearer without changing the meaning.

[revised manuscript text omitted]
 = 35\,\text{GHz}$ | Radar reflectivity factor | 3.5 s | $150 - 13000\,\text{m}$ | 30 m |
| | | Vertical velocity | 3.5 s | $150 - 13000\,\text{m}$ | 30 m |
| | | Linear depolarization ratio | 3.5 s | $150 - 13000\,\text{m}$ | 30 m |
| Raman-Polarization Lidar Polly$^{XT}$ (Engelmann et al., 2016) | $\lambda = 355, 532, 1064\,\text{nm}$ | Attenuated backscatter coeff. | 30 s | $100 - 15000\,\text{m}$ | 7.5 m |
| | $\lambda = 355, 532\,\text{nm}$ | Raman backscatter signal | 1 h | $300 - 5000\,\text{m}$ | $\sim 50\,\text{m}$ |
| | $\lambda = 355, 532\,\text{nm}$ | Linear depolarization ratio | 30 s | $100 - 15000\,\text{m}$ | 7.5 m |
| Microwave radiometer RPG HATPRO-G2 (Rose et al., 2005) | $\nu = 22.24 - 31.4\,\text{GHz}$ | Brightness temperatures | 1 s | column integral | |
| | $\nu = 51.0 - 58.0\,\text{GHz}$ | Brightness temperatures | 1 s | column integral | |
| Doppler Lidar HALO Photonics Streamline XR (Pearson et al., 2009) | $\lambda = 1.5\,\mu\text{m}$ | Attenuated backscatter coeff. | 2 s | $48 - 12000\,\text{m}$ | 48 m |
| | | Vertical velocity | 2 s | $48 - 12000\,\text{m}$ | 48 m |
| Ceilometer Jenoptik CHM15kx | $\lambda = 1064\,\text{nm}$ | Attenuated backscatter coeff. | 30 s | $15 - 15300\,\text{
[revised manuscript text omitted]